# Imaging of Insect Hole in Living Tree Trunk Based on Joint Driven Algorithm of Electromagnetic Inverse Scattering

**DOI:** 10.3390/s22249840

**Published:** 2022-12-14

**Authors:** Jiayin Song, Jie Shi, Hongwei Zhou, Wenlong Song, Hongju Zhou, Yue Zhao

**Affiliations:** Department of Mechanical and Electrical Engineering, Northeast Forestry University, Harbin 150040, China

**Keywords:** non-invasive, convolutional neural network, electromagnetic inverse scattering

## Abstract

Trunk pests have always been one of the most important species of tree pests. Trees eroded by trunk pests will be blocked in the transport of nutrients and water and will wither and die or be broken by strong winds. Most pests are social and distributed in the form of communities inside trees. However, it is difficult to know from the outside if a tree is infected inside. A new method for the non-invasive detecting of tree interiors is proposed to identify trees eroded by trunk pests. The method is based on electromagnetic inverse scattering. The scattered field data are obtained by an electromagnetic wave receiver. A Joint-Driven algorithm is proposed to realize the electromagnetic scattered data imaging to determine the extent and location of pest erosion of the trunk. This imaging method can effectively solve the problem of unclear imaging in the xylem of living trees due to the small area of the pest community. The Joint-Driven algorithm proposed by our group can achieve accurate imaging with a ratio of pest community radius to live tree radius equal to 1:60 under the condition of noise doping. The Joint-Driven algorithm proposed in this paper reduces the time cost and computational complexity of tree internal defect detection and improves the clarity and accuracy of tree internal defect inversion images.

## 1. Introduction

According to the Global Forest Resources Assessment report released by FAO in 2020, there are 4.06 billion hectares of forest in the world and, since 2015, an average of 10 million hectares of forestland has disappeared every year. The annual outbreak of forest pests will damage about 35 million hectares of forests. Some insects eat leaves and branches, but some trunk-boring pests such as red fat sacs live in the trunk and cannot be found in a timely manner, causing the deaths of trees and economic losses. Therefore, it is necessary to detect pest communities in the xylems of living trees in order to timely prevent and control pests and diseases and protect forest resources.

Mainstream methods for the detection of internal defects in living tree trunks include stress wave, ultrasonic, and computed tomography (CT) scanning [1,2,3]. However, most of these methods have their drawbacks [4,5]. For example, the stress wave method involves nailing into the trunk of the tree due to its detection characteristics [6]. This detection method will cause damage to the tree and cannot be non-invasive. The ultrasonic inspection process is susceptible to interference from the external environment, and coupling agents may cause environmental pollution [7]. CT equipment is costly and can easily cause radiation hazards to researchers [8,9]. Although nuclear magnetic resonance imaging has proven to be one of the most accurate and powerful characterization techniques in applications such as the diagnosis of wounded branch structures in beech trees [10], its high cost makes it unfeasible in most applications.

At present, electromagnetic inverse scattering is a good method for non-invasive testing. Electromagnetic inverse scattering (EMIS) [11] refers to the use of measured field data to invert the electrical property parameters of a scatterer in the detection region. A large body of research has been accumulated on the EMIS problem, which can be broadly classified into two categories. One type is the linearization of nonlinear problems, represented by the Born and Rytov approximations [12,13], which have very good imaging results for Low dielectric constants. Another type of inversion method is the nonlinear method [14], which uses an iterative idea to improve the inversion accuracy through continuous iteration; the main methods are the Contrast Source Inversion (CSI) and the Subspace Optimization Method (SOM) [15,16]. Although these nonlinear iterative methods improve accuracy compared to linear approximation methods, they have disadvantages such as sensitivity to initial values and slow convergence [17]. See Table 1 for details.

Although electromagnetic inverse scattering has a wide range of applications in engineering measurement and medical fields [18,19], when it is used to detect the interior of trees we find that, with larger trees, the relatively smaller proportion of the pest community area will make inversion imaging more difficult. This is because the electromagnetic inverse scattering is nonlinear, which easily makes the program fall into an error solution that differs from the accurate algorithm. Moreover, the scattering equation is highly ill-conditioned; that is, a small error in the input data may cause a great change in the output results. The traditional methods, whether iterative or non-iterative, do not have a very strong data processing capacity, so retrieving the model with a small proportion of pest communities is difficult.

The Deep Convolutional network is improved by studying the Contrast Source Inversion algorithm, and the Super-Resolution network is subsequently introduced. A Joint-driven Super-Resolution algorithm with a low computational load and high resolution is proposed to solve the smaller proportion of pest communities in living trees.

## 2. Electromagnetic Inverse Scattering Formulation

The algorithm is tested from a two-dimensional (2-D) perspective [20], and Figure 1 shows the model legend of our EMIS, constructed as a living tree–pest model. Our group will start with Maxwell’s equations [21] by constructing scattering theoretical equations to obtain the parameter optimization ideas. To find the solution of the integral equations and investigate two major types of model-driven and data-driven inversion algorithms, our group will add Super-Resolution networks based on issues such as noise clutter during imaging to form a model-driven deep learning super-resolution-based inversion algorithm [22,23,24,25].

In this paper, we use a two-dimensional plane wave as the incident wave to detect the unknown domain, denoted as $E^{s}$, and the scattered field, denoted as $E^{i}.$ The forward propagation of electromagnetic waves is composed of two equations known as the Lippmann-Schwinger equation [26]. The first is the equation of state, which describes the interaction of the wave scatters:(1)$$E^{t}\left(r\right)=E^{i}\left(r\right)+i\omega \mu_{0}\underset{D}{\int }G\left(r,r^{\prime }\right)\left[-i\omega \varepsilon_{0}\left(\varepsilon_{r}\left(r^{\prime }\right)-1\right)E^{t}\left(r^{\prime }\right)\right]dr^{\prime },$$
where ω=2πf is the angular frequency, $\mu_{0}$ is the air magnetic permeability, and $G\left(r,r^{\prime }\right)$ denotes the Green’s function. For a two-dimensional plane wave, the $G\left(r,r^{\prime }\right)$ is defined as
(2)$$G\left(r,r^{\prime }\right)=-\frac{j}{4}H_{0}^{\left(2\right)}\left(k_{0}\left|r-r^{\prime }\right|\right),$$
$\mathrm{where}\;H_{0}^{\left(2\right)}$ denotes the second class of Hankel functions [27], $k_{0}$ denotes the number of waves in free space, $r^{\prime }=\left(x^{\prime },y^{\prime }\right)$ is the source point of the object domain D,r=(x,y) denotes the position vector at the receiver, $E^{t}$ denotes the total field, expressed as $E^{t}=E^{s}+E^{i}$, and $\varepsilon_{r}\left(r^{\prime }\right)$ is the relative permittivity of the target scatter.

The second equation, known as the data equation, describes the equivalent current radiation process for the scattered field:(3)$$E^{s}\left(r\right)=i\omega \mu_{0}\underset{D}{\int }G\left(r,r^{\prime }\right)\left[-i\omega \varepsilon_{0}\left(\varepsilon_{r}\left(r^{\prime }\right)-1\right)E^{t}\left(r^{\prime }\right)\right]dr^{\prime }.\;$$

In (3), the physical meaning of $-i\omega \varepsilon_{0}\left(\varepsilon_{r}\left(r^{\prime }\right)-1\right)$ is the inductive contrast current density, but we define the standardized contrast current density as $J\left(r\right)=\left(\varepsilon_{r}\left(r\right)-1\right)E^{t}\left(r\right)$, making variable $k_{0}=\omega \sqrt{\mu_{0}\varepsilon_{0}}$, express $\varepsilon_{r}\left(r\right)-1$ as contrast χ(r), and introducing operators $G_{s}\left(\cdot \right)$ and $G_{D}\left(\cdot \right)$.
(4)$$k_{0}^{2}\underset{D}{\int }G\left(r,r^{\prime }\right)J\left(r^{\prime }\right)dr^{\prime }=\{\begin{array}{c} G_{s}\left(J\right),r\in S\\ G_{D}\left(J\right),r\in D \end{array}.\;$$

The control equations can then be written in two non-permissible forms.

The first is called the field-type equation, where the electric field involves two equations:(5)$$E^{t}\left(r\right)=E^{i}\left(r\right)+G_{D}\left(\chi E^{t}\right),r\in D,$$
(6)$$E^{s}\left(r\right)=G_{s}\left(\chi E^{t}\right),r\in S.$$

The second is called the source-type equation, in which the current involves two equations:(7)$$J\left(r\right)=\chi \left(r\right)\left[E^{i}\left(r\right)+G_{D}\left(J\right)\right],r\in D,$$
(8)$$E^{s}\left(r\right)=G_{s}\left(J\right),r\in S.\;$$

The electromagnetic inverse scattering imaging algorithm is based on the above two types of equations to solve for the target cross-section parameters.

## 3. Joint-Driven Algorithm

To overcome the nonlinearity and high morbidity of inverse scattering and to solve the problem that the pest community accounts for a small proportion of the cross-sectional area of living trees, a Joint-Driven algorithm is proposed by combining the data-driven algorithm with the model-driven algorithm. The advantage of the model-driven algorithm in solving problems lies in accurate mathematical modeling based on the mechanism information of the problem itself, while the disadvantage is its excessive dependence on the initial value. The data-driven algorithm relies on the extraction of feature information in the data to give full play to the advantages of big data. It can only solve problems by constantly learning the relationship between data. Therefore, this paper introduces the model-driven mechanism into the data-driven, integrating the advantages of both.

Because the non-destructive testing of trees using electromagnetic waves is almost carried out in the field, it causes the data we obtain to contain noise. To improve the resolution, increase the clarity, and optimize the inversion results, we add a super-resolution network after the inversion and train it in a targeted way so that it can match the optimized inversion results. We call this behavior the Joint-Driven Super-Resolution algorithm.

Neural networks are typically data-driven algorithms, with Deep Convolutional neural networks being the most capable of extracting and classifying data features [28]. The convolutional neural network structure in this paper is shown in Figure 2 and is a typical ConvNets network, consisting of four types of layers: input layer, convolutional layer, pooling layer, and fully connected layer.

$E^{s}$ is fed into the network as a 32 × 32 matrix form as an input into the convolutional network. The perceptual field size in the network is 5 × 5, and the feature data is evaluated in the pooling layer with a 2 × 2 pooling operation. Finally, a 100 × 100 regression value is involved as the output for imaging through the fully connected layer.

### 3.1. Unite the Contrast Source Inversion with a Deep Convolutional Network

To prevent the occurrence of the overfitting phenomenon in the process of neural network training—that is, when the neural network has achieved good detection accuracy on the training data, but the detection accuracy on the test data set is greatly decreased–it is necessary to normalize the cost function. In this paper, L2 normalization is adopted to improve the cost function.
(9)$$C=-\frac{1}{n}\underset{x_{j}}{\sum }\left[y_{j}\ln a_{j}^{L}+\left(1-y_{j}\right)\ln\left(1-a_{j}^{L}\right)\right]+\frac{\lambda }{2n}\underset{\omega }{\sum }\omega^{2},$$

If $\frac{\lambda }{2n}\underset{\omega }{\sum }\omega^{2}$ is the normalization term and the cost function before improvement is defined as $C_{0}$, then the cost function after specification can be written as
(10)$$C=C_{0}+\frac{\lambda }{2n}\underset{\omega }{\sum }\omega^{2}.$$

The partial derivative of C concerning the weight and bias in the network is
(11)$$\frac{\partial C}{\partial \omega }=\frac{\partial C_{0}}{\partial \omega }+\frac{\lambda }{n}\omega ,$$
(12)$$\frac{\partial C}{\partial b}=\frac{\partial C_{0}}{\partial b}.$$

In (12), it shows that the partial derivative of the cost function L2 concerning bias does not change after specification and the biased gradient descent rule does not change, while the weight learning rule becomes
(13)$$\omega \to \left(1-\frac{\eta }{m}\right)\omega -\frac{\eta }{m}\underset{x}{\sum }\frac{\partial C_{x}}{\partial \omega }.$$

In the learning process, the weight is adjusted by the factor $\left(1-\frac{\eta }{m}\right)$.

In the cost function, the learning rate η and parameter λ are hyperparameters. Although they are not directly related to the establishment of the Deep Convolutional neural network, the selection of hyperparameters λ and η has a great relationship with the training speed and imaging effect.

To speed up the training process of a Deep Convolutional neural network and optimize the selection of hyperparameters, the Contrast Source Inversion is combined with the cost function of a Deep Convolutional neural network. The core of the Contrast Source Inversion is to minimize the objective function [29,30], which is in the form of
(14)$$F\left(J_{j},\chi ,\beta \right)=\beta \frac{\sum_{j}∥\chi E_{j}^{inc}-J_{j}+\chi G_{D}J_{j}{∥}_{D}^{2}}{\sum_{j}∥\chi E_{j}^{inc}{∥}_{D}^{2}}+\frac{\sum_{j}∥E_{j}^{s}-G_{S}J_{j}{∥}_{S}^{2}}{\sum_{j}∥E_{j}^{s}{∥}_{S}^{2}}.$$

The aim of the Deep Convolutional neural network is to acquire the appropriate weight and bias by minimizing the cost function, so the normalization of the Contrast Source Inversion is introduced into the Deep Convolutional neural network and the value of the hyperparameter λ is determined by the mechanism of information of the Contrast Source Inversion. Formally, the value of λ is changed from a constant value to a dynamic real value determined by the Contrast Source Inversion to form a Jointly-Driven deep learning network. Therefore, the cost–function form of a model-driven deep learning network is:(15)$$C=-\frac{1}{n}\Sigma_{x_{j}}\frac{\left[y_{j}\ln a_{j}^{L}+\left(1-y_{j}\right)\ln\left(1-a_{j}^{L}\right)\right]}{\Sigma_{j}\Sigma_{k}∥E_{k}^{s}{∥}_{s}^{2}}+\frac{\lambda_{0}}{2n\Sigma_{j}\Sigma_{k}∥E_{k}^{s}{∥}_{s}^{2}}+\Sigma_{\omega }\omega^{2}.$$

In (15), $\Sigma_{j}\Sigma_{k}∥E_{k}^{s}{∥}_{s}^{2}$ represents the j group of data and the sum of the binary norm of scattering field data obtained by K receivers in each group of data. Although $E_{k}^{s}$ is known, $E_{k}^{s}$ in each group of training data is not equal, so $\Sigma_{j}\Sigma_{k}∥E_{k}^{s}{∥}_{s}^{2}$ is a dynamic real value. $\lambda_{0}$ is a constant weighting coefficient. In addition to preventing the Deep Convolutional neural network from over-fitting, $\lambda_{0}$ can also prevent abnormal training data input from causing $\Sigma_{j}\Sigma_{k}∥E_{k}^{s}{∥}_{s}^{2}$ mutation, thus reducing the training effect of Deep Convolutional neural network.

### 3.2. Analysis and Optimization of Weights

To speed up training, weights and bias are optimized. Traditional convolutional networks are selected by Gaussian random variables, and the mean of normalized Gaussian distribution is 0 and the standard deviation is 1 [31]. Since z=∑ωx+b, z itself obeys the Gaussian distribution after initializing ω and b using gaussian random variables, as shown in Figure 3.

As can be seen in Figure 3, the slope of the curve is small and the overly large input values make σ(z), the output data, almost saturated. This characteristic makes the weight updating process change very slowly. To improve the learning speed of the network, this paper sets the mean of the Gaussian random distribution to 0 and the standard deviation to 1nin, where $n_{in}$ is the number of input neurons. The improved z distribution is shown in Figure 4. 

### 3.3. Super-Resolution Network-Assisted Imaging

When the proportion of pest communities in the cross-section of trees is less than 10%, the inversion results will be noisy. If the proportion of the pest community in the cross-section of the tree becomes smaller and smaller, the noise of the inversion image becomes denser and denser. To solve this problem, an image optimization and image reconstruction technology is introduced: Real-ESRGAN [32]. The Super-Resolution algorithm can not only improve the resolution of the inversion image but also optimize the image quality and remove the dryness of the image. Real-ESRGAN uses a U-Net discriminator with spectral normalization [33,34,35], which can identify where the noise is and where the pest community is located. Its network structure is shown in Figure 5.

The U-Net discriminator focuses not only on the image as a whole but also on the characteristics of each part, the most important for inverse imaging being the shape and position of the scatter.

### 3.4. Evaluation Indicators

The results are evaluated using Mean Square Error (MSE) and Image Intersection Over Union (*IOU*) as a criterion for the accuracy of our single inversion images [36], where IOU is defined as
(16)$$IOU=\frac{S_{i}\cap S_{f}}{S_{i}\cup S_{f}}.$$

$S_{i}$ is the area of internal pests in the scatterer in the inversion diagram and $S_{f}$ is the area of the internal pest community set in the living tree–borers model. It ·is specified that a single pest detection is judged to be accurate when IOU>0.87; ideally, IOU=1. To test the performance of the algorithm, this paper sets up multiple groups of experimental data to test the algorithm and calculates the accuracy rate of the algorithm in different detection environments. The accuracy rate formula is as follows.
(17)$$A_{cc}=\frac{N_{tp}}{N_{t}}\times 100\%.$$

$A_{cc}$ represents the detection accuracy of the algorithm for all test data, $N_{tp}$ is the number of all test results that are judged to be accurate, and $N_{t}$ is the total number of test data.

## 4. Experiments and Results Analysis

### 4.1. Experiment Condition

Considering that the detection of trees is mostly carried out outdoors, in order to better simulate the noise interference in the field detection in real life, a method of artificially adding interference factors was proposed to simulate the detection of trees in the field. The noise construction equation is shown below.
(18)$$E_{n}=\frac{1}{\sqrt{2}}\sqrt{\frac{1}{N_{s}N_{i}}}E_{0}^{s}n_{0},$$
where $E_{n}$ is our constructed noise scattered field data, $E_{0}^{s}$ is the scattered field data received by the receiver, $N_{i}$ is the number of transmitters,$N_{s}$ is the number of receivers, and $n_{0}$ is the noise matrix, which is structured as follows:(19)$$n_{0}=n_{l}\left(\left[\begin{array}{ccc} 1+i & \cdots & 1+i\\ \vdots & \ddots & \vdots \\ 1+i & \cdots & 1+i \end{array}\right]\right),$$
$\mathrm{where}\;n_{l}$ is the noise factor, which can be changed by changing the value of $n_{l}$, and $n_{0}$ is the $N_{i}\times N_{s}$ matrix. Our final input scattered field data for training $E^{s}$ is
(20)$$E^{s}=E_{0}^{s}+E_{n}.$$

To better describe the ratio of pest community size to the cross-sectional area of wood, we call it the Inversion Ratio, and its formula is as follows:(21)$$Inversion\;Ratio=\frac{r_{p}}{r_{t}}.$$
$\mathrm{where}\;r_{p}$ represents the radius of the pest community in m and $r_{t}$ is the radius of the living tree in meters.

This section uses simulated electromagnetic inverse scattering data to validate our proposed method with the simulation model parameters set as shown in the Table 2 below.

Our group used finite element simulations to obtain 18,000 sets of scattering data in each of the two states. To ensure generalization, we selected a random sample of 15,000 sets of data in the dataset to train our proposed combination of model-driven and data-driven methods. Another 500 sets of data in the remaining 3000 sets were randomly selected for testing.

Within 2 m × 2 m square areas, white indicates air; brown is a two-dimensional cross-section of the xylem, a circle of radius 0.6 m with a relative permittivity of 7; and black indicates an internal trunk pest community with a relative permittivity of 60.

### 4.2. Imaging Experiment of Insect Hole in Living Trees

#### 4.2.1. Relationship between Accuracy and InversionRatio

Using the Deep Convolutional algorithm, experiments were carried out on four models with different sizes but the same *Inversion Ratio*. The results are shown in Figure 6.

The accuracy of the inversion results was evaluated according to Formula (16). The results are shown in Table 3, which demonstrates that under the condition of maintaining the Inversion Ratio, changing the radius will not affect the accuracy value.

#### 4.2.2. Detection of Pest Communities with Different InversionRatio Values

The Contrast source Inversion, Deep Convolution algorithm, Joint-Driven algorithm, and Joint-Driven Super-Resolution algorithm were used to detect pest communities with different Inversion Ratios. The results are shown in Figure 7 and Figure 8. The IOU is calculated according to Formula (16) to evaluate the accuracy of inversion results, and the results are shown in Table 4.

Figure 7, Figure 8 and Figure 9 respectively calculate the IOU of six different Inversion Ratio inversion results by the four algorithms, and the values are shown in Table 4.

### 4.3. Discussion of Experimental Results

When Inversion Ratio=1:60, the Contrast Source Inversion cannot reverse a regular graph, resulting in IOU = error. When the inversion ratio was 1:30 to 1:50, the pest community size retrieved by the Contrast Source Inversion algorithm was the same. When Inversion Ratio=1:10 and Inversion Ratio=1:20, the Contrast Source Inversion algorithm imaging produces a certain “halo.” With a threshold value of 80% of the highest relative dielectric constant value in the imaging image, only the red part of the image can be observed.

Regarding the Deep Convolutional algorithm: when Inversion Ratio=1:20, 1:30, and 1:40, its inversion insect pest community size is unchanged, and its algorithm is insensitive to size change, which is prone to error.

When the Joint-Driven algorithm inverts the model of Inversion Ratio=1:10, 1:20, the inversion results are clear and accurate. Moreover, the Joint-Driven algorithm can accurately reverse the size of the pest community. Although there are interference points in the inversion results, the location of the pest community can also be distinguished.

The Joint-Driven Super-Resolution algorithm can not only de-noise inverted images, but also further highlights the position of the pest community, further improves the inversion effect, and is more conducive to judging the internal conditions of trees by further optimizing the image.

Figure 10 demonstrates the Inversion Ratio=1:30 single–partial pest detection process because when the Inversion Ratio is between 1:30 and 1:60, Contrast Source Inversion cannot correctly show the shape of the scatter. The Contrast Source Inversion, Deep Convolution, and Joint-Driven methods are compared to analyze the stability of model-driven deep learning networks in the iterative process.

Figure 10a shows that the Contrast Source Inversion tends to stabilize after around 500 iterations, with the number of iterations being higher and more time-consuming. Figure 10b shows that the Deep Convolutional algorithm needs to iterate about 350 times to reach the stable range, and the training process is not smooth enough. Figure 10c shows that the Joint-Driven algorithm only needs about 60 iterations to reach the stable state, which greatly reduces the number of iterations, improves the imaging time, and smooths the training process.

According to Equation (17), the detection accuracy statistics of each inversion algorithm were carried out separately for each of the three radii, and Figure 11 shows the detection accuracy of each algorithm in 300 sets of single-pest inversion tests.

As can be seen from Figure 11, the Contrast Source Inversion can only invert the living tree burrow scatters with an Inversion Ratio greater than 1:30 and its application range is limited. Although the Deep Convolutional network algorithm can invert living tree burrow scatters with an Inversion Ratio=1:60, its accuracy is only about 70%, which cannot meet the actual requirements. The Joint-Driven algorithm proposed in this paper not only solves the problem of imaging the wood cross-sectional area with a small insect colony proportion, but it also has an inversion accuracy of up to 90% after the optimization of a specially-trained super-resolution network.

To sum up, the four methods used are summarized and the results are shown in the following Table 5.

When Inversion Ratio=1:10 and Inversion Ratio=1:20, the Contrast Source Inversion algorithm can reverse meaningful results. If the Inversion Ratio continues to change, the results generated by the Contrast Source Inversion algorithm will lose their reference value. Therefore, the calculation error is represented by ‘Non’.

## 5. Conclusions

In this paper, a model-driven deep learning Super-Resolution inversion algorithm is proposed to solve the problem of high noise and poor imaging in electromagnetic wave detection of tree pest communities. By studying the propagation process of electromagnetic waves in scatters and combining the advantages of the Contrast Source Inversion, Deep Convolutional network, and Super-Resolution network, a Joint-Driven Super-Resolution algorithm is proposed. This algorithm overcomes the problem that the selection of neural network structure and the process of parameter optimization depend too much on the experience of the experimenter by introducing the Contrast Source Inversion, so that the neural network can better fit the nonlinear problem and overcome the ill-posed problem by learning a large amount of data. The experiments are carried out by continuously reducing the radius of the model pest community and using the Contrast Source Inversion, the Convolutional Network algorithm, the Joint-Driven algorithm, and the Joint-Driven Super-Resolution algorithm. Our solution to the problem of imaging tiny high-contrast scatters is provided.

## Figures and Tables

**Figure 1 sensors-22-09840-f001:**
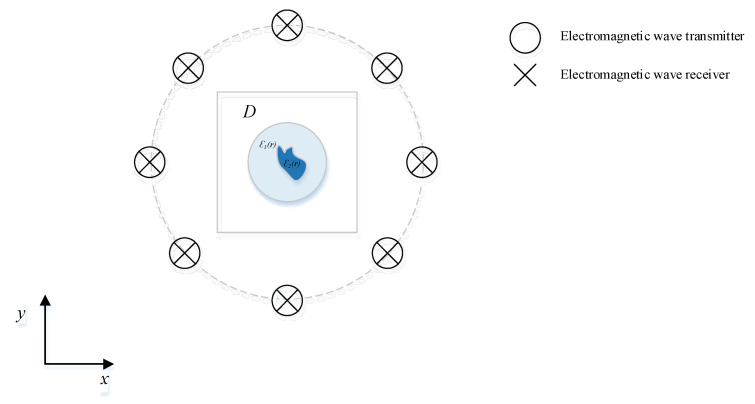
Electromagnetic inverse scattering model.

**Figure 2 sensors-22-09840-f002:**
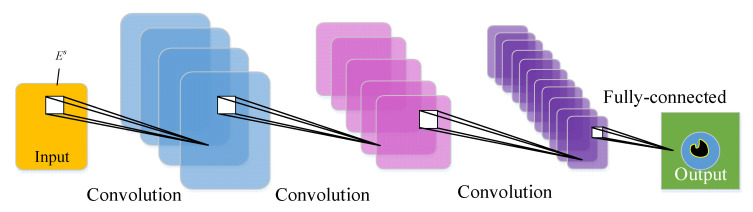
Structure of the convolutional neural network in this paper.

**Figure 3 sensors-22-09840-f003:**
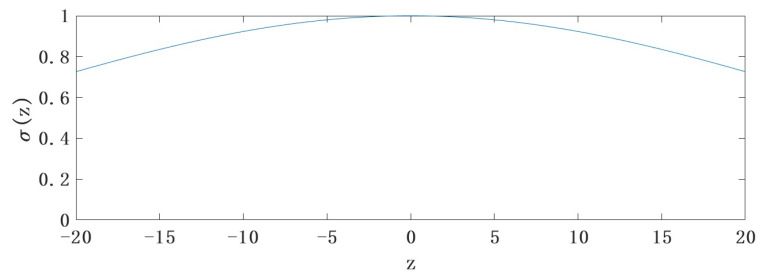
Gaussian distribution of z.

**Figure 4 sensors-22-09840-f004:**
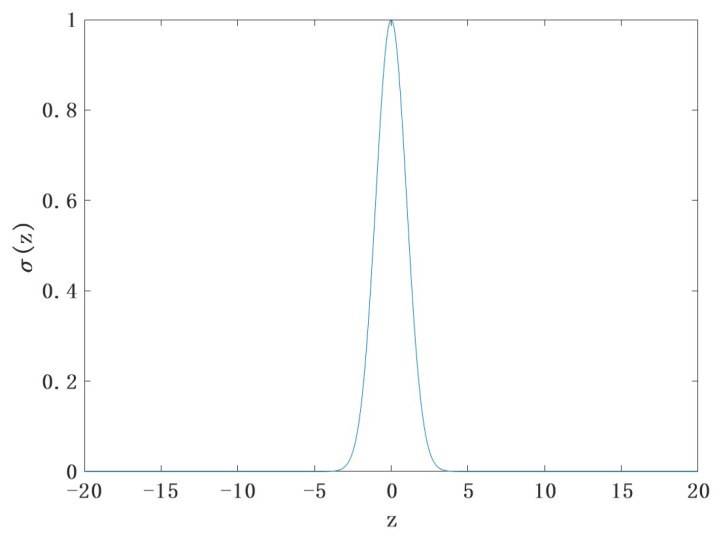
Gaussian distribution of z after improvement.

**Figure 5 sensors-22-09840-f005:**
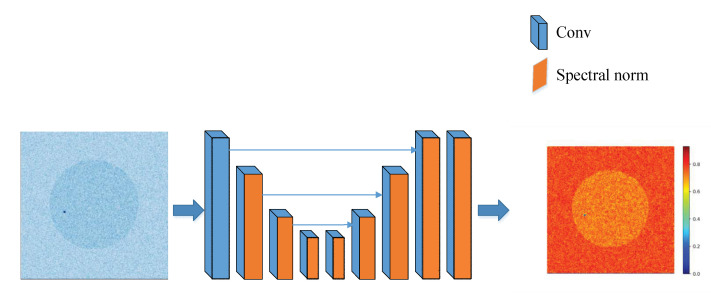
The architecture of the U-Net discriminator with spectral normalization.

**Figure 6 sensors-22-09840-f006:**
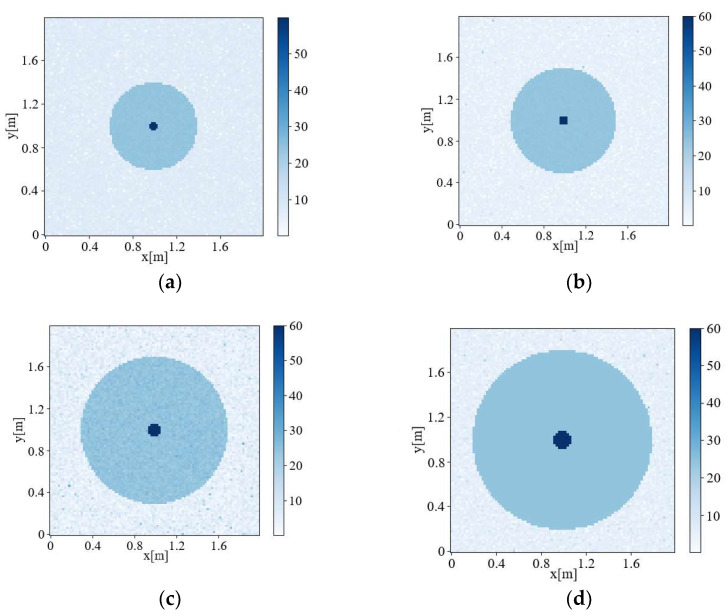
Inversion images with the same inversion ratio and different sizes: (**a**) 0.04 m radius of pest community; 0.4 m radius of the living tree, (**b**) 0.05 m radius of pest community; 0.5 m radius of the living tree, (**c**). 0.07 m radius of pest community; 0.7 m radius of the living tree, and (**d**) 0.08 m radius of pest community; 0.8 m radius of the living tree.

**Figure 7 sensors-22-09840-f007:**
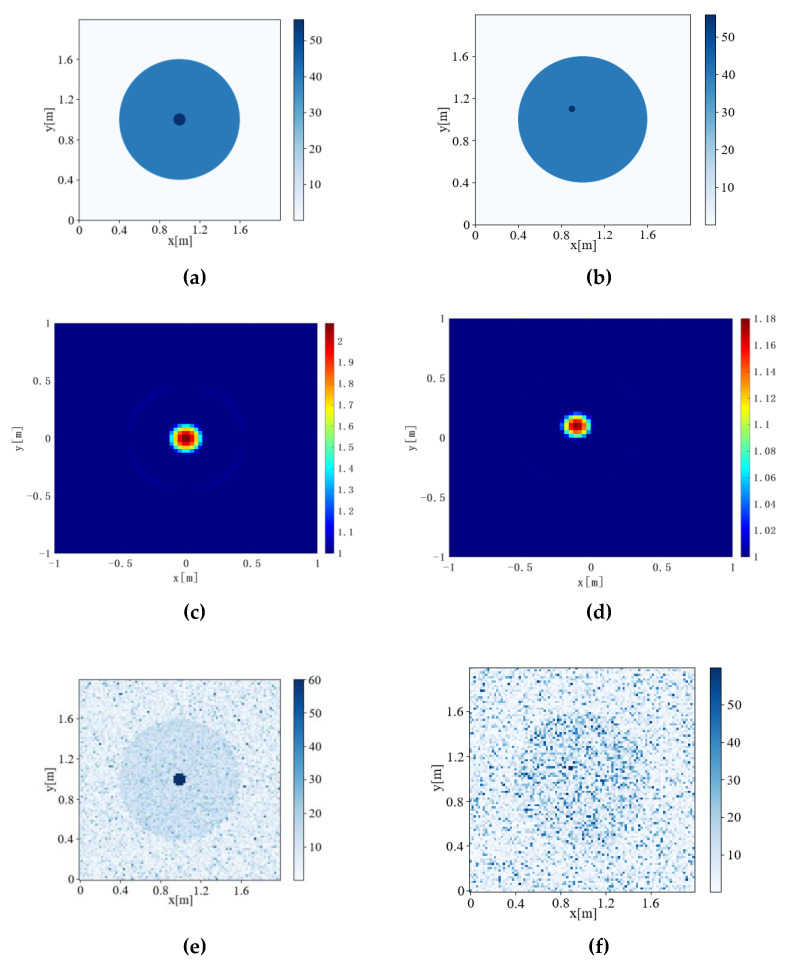
Inversion results of the Inversion Ratio from 1:10 to 1:20. (a) Model diagram of Inversion Ratio = 1:10, (b) model diagram of Inversion Ratio = 1:20, (c) Contrast Source Inversion results of Inversion Ratio = 1:10, (d) Contrast Source Inversion results of Inversion Ratio = 1:20, (e) Deep Convolutional Inversion results of Inversion Ratio = 1:10, (f) Deep Convolutional Inversion results of Inversion Ratio = 1:20, (g) joint-Driven inversion results of Inversion Ratio = 1:10, (h) joint-Driven inversion results of Inversion Ratio = 1:20, (i) Joint-Driven super-resolution inversion results of Inversion Ratio = 1:10, and (j) Joint-Driven super-resolution inversion results of Inversion Ratio = 1:20.

**Figure 8 sensors-22-09840-f008:**
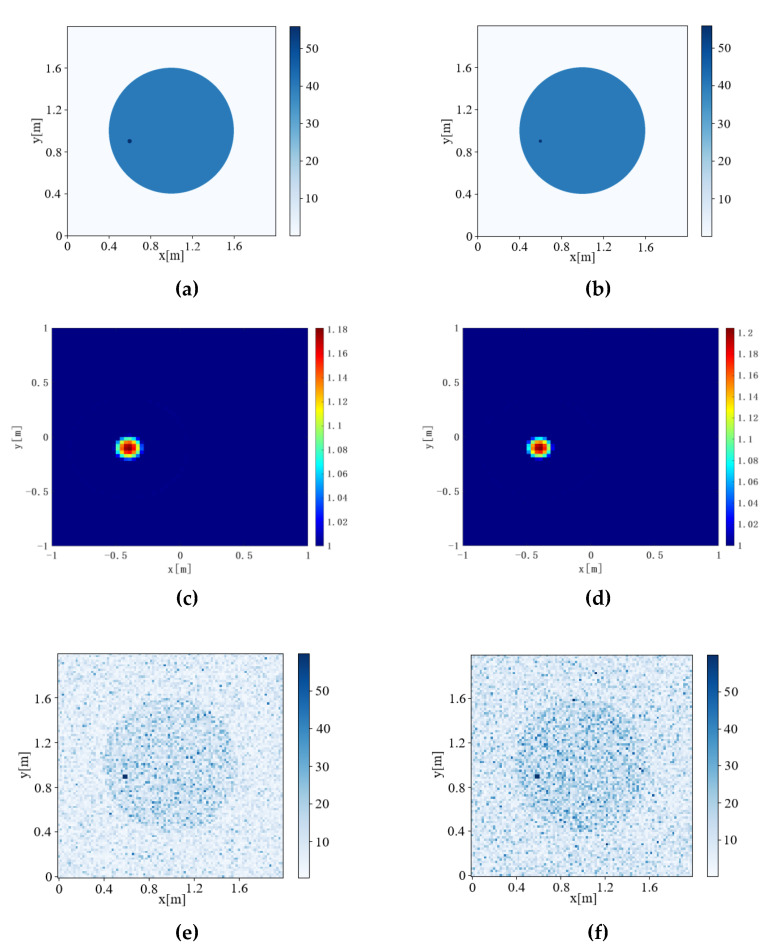
Inversion results of Inversion Ratio from 1:30 to1:40. (a) Model diagram of Inversion Ratio = 1:30, (b) model diagram of Inversion Ratio = 1:40, (c) Contrast Source Inversion results of Inversion Ratio = 1:30, (d) Contrast Source Inversion results of Inversion Ratio = 1:40, (e) Deep Convolutional Inversion results of Inversion Ratio = 1:30, (f) Deep Convolutional Inversion results of Inversion Ratio = 1:40, (g) Joint-Driven inversion results of Inversion Ratio = 1:30, (h) Joint driven inversion results of Inversion Ratio = 1:40, (i) Joint-Driven Super-resolution inversion results of Inversion Ratio = 1:30, and (j) Joint-Driven Super-resolution inversion results of Inversion Ratio = 1:40.

**Figure 9 sensors-22-09840-f009:**
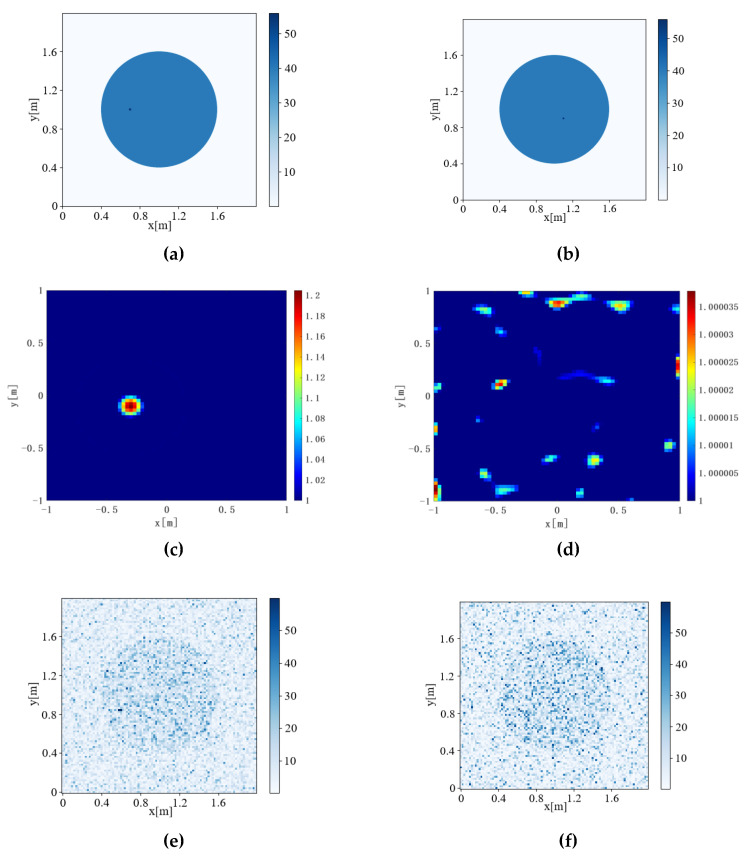
Inversion results of Inversion Ratio from 1:50 to 1:60. (a) Model diagram of Inversion Ratio = 1:50. (b) Model diagram of Inversion Ratio = 1:60. (c) Contrast Source Inversion results of Inversion Ratio = 1:50. (d) Contrast Source Inversion results of Inversion Ratio = 1:60. (e) Deep Convolutional Inversion results of Inversion Ratio = 1:50. (f) Deep Convolutional Inversion results of Inversion Ratio = 1:60. (g) Joint driven inversion results of Inversion Ratio =1:50 (h) Joint driven inversion results of Inversion Ratio =1:60 (i) Joint-Driven Super-Resolution inversion results of Inversion Ratio = 1:50. (j) Joint-Driven Super-Resolution inversion results of Inversion Ratio = 1:60.

**Figure 10 sensors-22-09840-f010:**
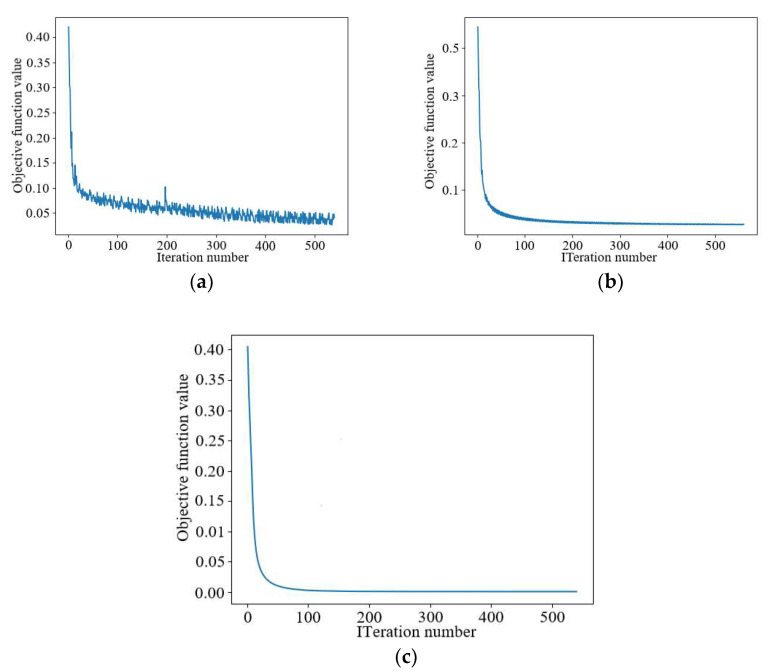
Number of algorithm iterations versus MSE. (**a**) Contrast Source Inversion iteration process, (**b**) Deep Convolutional network training process, and (**c**) the Joint-Driven algorithm trains the iterative process.

**Figure 11 sensors-22-09840-f011:**
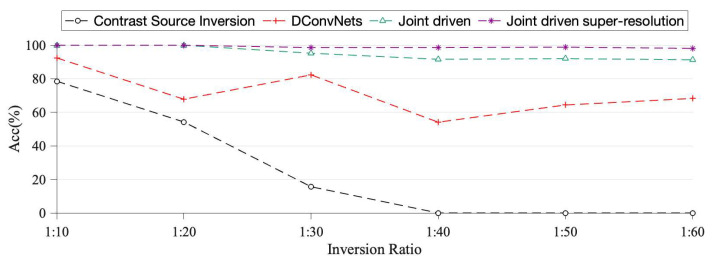
Accuracy of four inversion algorithms for detection.

**Table 1 sensors-22-09840-t001:** Summary of the advantages and disadvantages of algorithms.

Algorithm	Algorithm Defects	Algorithm Advantages
CSI	Sensitive initial value; slow convergence speed; unable to process large-scale data	Iterative solution; not involving solving the positive problem
SOM	Sensitive initial value; unable to process large-scale data; large amount of calculation	Reduces CSI solving dimension; improves solving speed and success probability
CNN	Needs a lot of data training; shifts the computational burden to the learning stage	No physical modeling required

**Table 2 sensors-22-09840-t002:** Simulation parameter settings.

Parameter Name	Parameter Values	Parameter Name	Parameter Values
Domain of solution	2 m × 2 m	The relative permittivity of pest community	60
The radius of a living tree	0.6 m	Air resistance	120π
Inversion Ratio	1:10~1:60	Number of electromagnetic emitters	32
Electromagnetic frequency	200 MHz~700 MHz	Number of electromagnetic wave receivers	32
The relative permittivity of air	1	The internal relative dielectric constant of tree	7
Noise factor $n_{l}$	0.2		

**Table 3 sensors-22-09840-t003:** IOU of each Inversion image when Inversion Ratio =1:10.

	Figure 6a	Figure 6b	Figure 6c	Figure 6d
IOU	0.976	0.975	0.977	0.979

**Table 4 sensors-22-09840-t004:** IOU of each algorithm.

	Contrast Source Inversion	Deep Convolutional Inversion	Joint Driven Inversion	Joint-Driven Super-Resolution Inversion
1:10	0.885	0.954	1	1
1:20	0.757	0.775	1	1
1:30	0.541	0.851	0.955	0.984
1:40	0.432	0.653	0.953	0.982
1:50	0.325	0.773	0.958	0.988
1:60	error	0.763	0.954	0.986

**Table 5 sensors-22-09840-t005:** Comparison by the Four Methods.

			Contrast Source Inversion	Deep Convolutional Inversion	Joint-Driven Inversion	Joint-Driven Super-Resolution Inversion
Iteration stability times			500	350	60	60
Inversion Ratio	1:10	Maximum error	11.1%	6.6%	3.3%	2.5%
		Minimum error	7.3%	3.6%	1.5%	1.3%
	1:20	Maximum error	23.3%	15.5%	7.2%	3.3%
		Minimum error	15.4%	10.2%	4.5%	1.5%
	1:30	Maximum error	Non	17.5%	8%	4.2%
		Minimum error	Non	14.3%	4.8%	2.3%
	1:40	Maximum error	Non	25.8%	8.6%	4.9%
		Minimum error	Non	22.3%	5.1%	2.8%
	1:50	Maximum error	Non	36.4%	9.2%	5.6%
		Minimum error	Non	28.5%	5.5%	3.7%
	1:60	Maximum error	Non	41.2%	10.1%	6.5%
		Minimum error	Non	33.3%	6.5%	4.3%

## Data Availability

Not applicable.

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
