# Peer review of "Imaging of Insect Hole in Living Tree Trunk Based on Joint Driven Algorithm of Electromagnetic Inverse Scattering"

_sensors, 2022, doi:10.3390/s22249840_

Round 1

Reviewer 1 Report

This paper presents a reasonable method to solve a real application problem. It is well-organized, clearly writing, and shows some interesting results that encouraged to be accepted with minor revision. However, the commented questions need only to be answered.

1.    Please explicitly indicate and clarify the challenges this study aims to address. What are the challenges and why? Why cannot the previous studies well address these challenges. 

2.    At the end of section 1 add a table that summarizes the advantages and disadvantages of existing methods facing the same problem. This way the reader would rapidly appreciate novelty of the paper.

3. Please enrich the captions of all figures and tables for clarification.

4.  In the comparison to SOTA methods, more experimental results of other state-of-the-art methods should be given.  

5. I also find some grammar problems in this paper. Author needs to carefully check these low mistakes, which is very important for readers.

6.  Related to the author's data-driven algorithm, why does the author use a simple CONVNET algorithm with four types of layers? There are many types of deep networks used as data-driven including attention and transformers which are more popular and demonstrate better performance.

Reviewer 2 Report

In the paper the authors presented a new method for non-invasive detecting of tree interior in order to locate the trees eroded by trunk pests. The method based on electromagnetic inverse scattering was applied. To determine the extent and location of pest erosion of the trunk the joint-driven algorithm, which can effectively solve the problem of unclear imaging, was proposed. An interesting problem has been discussed, nonetheless there are some remarks:

1. The abbreviation of CT needs to be explain earlier, in line 34, not 39.

2. There are few sentences that are incomprehensible (e.g. lines 57-59, 162, 204, 205, 213, 220, 229, 243-244, 249-250, 286. Grammar mistakes, punctuation, misspellings need to be corrected. A thorough proofreading in necessary.

3. Equations (11) and (12) are the same. Please, verify this.

4. There is no 'a' in the formula (18), however there is 'a' in the explanation below the equation. Please, verify this.

5. The descriptions in the drawings are illegible. The font is too small. Please, correct this.

6. In my opinion, in Table 2, instead of 'Image a, b, c, d' there should be 'Image 6a, 6b, 6c, 6d'.

7. Line 224, 225 - please use 'meters' instead of 'm'.

8. Check the caption under Figures: 7h there should be 1:20, 7i - 1:30, 8 - 1:40.

9. The conclusions are cursory, only summarizing the techniques used. Please, emphasise the novelty and the advantages of the presented approach.

Reviewer 3 Report

This paper implements Implements the Image of an insect hole in the trunk of a tree electromagnetic inverse scattering The work done is interesting and meets the standards of the sensor journal, but needs a major revision before being published in the journal. 

Minor comments:

Include some more reference in the introduction on microwave tree trunk imaging algorithms.

Improve the quality of figures 3,4,6,7,8 and 10.

Specific comments:

Authors should include a table comparing the improvements of their algorithm with respect to others in terms of time and resolution. 

What is the maximum and minimum error made when using this algorithm?

This algorithm could be used at higher frequencies and in a different Er range, for example, if the holes were not air holes. 

Round 2

Reviewer 3 Report

The authors have responded to my suggestions and comments and the article is ready for publication in the journal.
